# Preparation of Modified Colloidal Gas Aphrons and Analysis of the Oil Displacement Effect

**Shaohua You, Xiaofei Sun \*  and Xiaoyu Li**

School of Petroleum Engineering, China University of Petroleum (East China), Qingdao 266580, China; 1702010730@s.upc.edu.cn (S.Y.); b19020028@s.upc.edu.cn (X.L.)

\* Correspondence: xfsun@upc.edu.cn

**Abstract:** Colloidal gas aphrons (CGAs) offer some advantages in improving oil recovery, but resin and asphaltene deposition problems still occur in CGA flooding. Based on this phenomenon, a new modified colloidal foam system is developed by incorporating a modifier in CGA preparation. The results indicate that the modified CGAs prepared by adding foaming agent sodium dodecyl sulfate (SDS) (concentration: 5 g/L) and GXJ-C (a CGAs modifier from a light fraction of petroleum; concentration: 0.1 g/L) attained the best performance. Oil displacement experiments show that modified CGA flooding had a better effect than water or CGA flooding. There are two important mechanisms via which modified CGAs enhance oil recovery, including decreasing the interfacial tension and enhancing the heavy components in the recovered oil. The developed modified CGA system attained a good oil displacement effect, which is of guiding significance to further improve the oil displacement efficiency and application of foam flooding.

**Keywords:** colloidal gas aphrons; modifier; oil displacement experiment; asphaltene

## 1. Introduction

Commonly used improved recovery methods in oil fields include chemical flooding, thermal oil recovery, and miscible flooding. Foam flooding, as a type of chemical flooding, is widely used in oil fields. Addition of a certain concentration of foaming agent into the water is supplemented by gas (air, natural gas, or flue gas) to form foam, and its high viscosity helps reduce the mobility ratio and increase the sweep efficiency of the injectant [1,2]. The mechanism of foam flooding can be divided into microcosmic and macroscopic oil displacement [3,4]. Under normal conditions, after entering the reservoir, the foaming agent firstly enters the large pore channels where the resistance may be higher than that in the small channels due to the gas blockage effect, which may lead to temporary blockages. Regarding the size difference of foam and its pseudoplastic effects, when the resistance in the macropores increases, resulting in a higher driving gradient pressure than that in the micropores, the water phase in the foam system may permeate the micropores, and the foam disintegrates. Starting from the frontal drive, the whole driving process alternately changes and, finally, an oil-rich zone is built up along the foam front, which is driven to the well [5]. At present, domestic and international researchers conducted a series of foam flooding field tests on the basis of inhouse laboratory investigations and attained certain effects. The displacement method not only greatly improves sweep efficiency, but also reduces the environmental damage commonly caused by chemical flooding, which is a promising recovery technology [6].

Compared with conventional colloidal liquid aphrons (CLAs) [7–11], colloidal gas aphrons (CGAs; first proposed in 1971) [2,5,12,13] prepared by high-speed stirring (faster than 8000 rpm) feature small particles (10–100 µm), a large specific surface area, and a relatively stable performance in surfactant solutions. The CGA consists of a bubble core and a soap film layer that envelops the gas phase.

The soap film includes inner and outer surfaces, and a surfactant monolayer is adsorbed onto both surfaces [14]. Compared with common foams, CGAs contain microbubbles and have a high gas bearing capacity, large specific surface area, good coalescence, dynamic stability, and viscosity and fluidity similar to water. The CGA consists of a gaseous core and a layer of soap-coated gaseous phase. The soap film is divided into internal and external surfaces. On each surface, a surfactant monolayer is adsorbed. Hydrophobic groups are directed outward, and hydrophilic groups are oriented toward the inside. However, at the interface between the gas and water phases, the hydrophobic groups are oriented toward the soap film, and the hydrophilic groups are oriented toward the bulk water phase, which results in double electric layers. Therefore, there is a large difference between natural and synthetic bubbles [9]. CGAs offer advantages in improving oil recovery, but resin and asphaltene deposition problems still occur in CGA flooding. Based on this issue, a new modified colloidal foam system is developed by introducing a modifier in CGA preparation, which provides an effective solution to the aforementioned problems and is of guiding significance for further improving the oil displacement efficiency of gas foam.

## 2. Method

### 2.1. Preparation of the Modified CGAs

Modified CGAs were prepared as follows: (1) 100 mL of the modified CGA system solution was prepared with 0.5 g of foaming agent and 0.01 g of modifier; (2) the prepared solution was poured into a homemade cylinder container to prepare modified CGAs at a specific concentration and a certain stirring rate, time, and temperature. Figure 1 shows a schematic diagram of the preparation device (the high-speed stirrer used in this study was a ZNGJ-2 model instrument, made by Qingdao Tongchun Oil Instrumengt Co., Ltd. with a digital display and stepless speed regulation, and the speed regulation range was 0–15,000 rpm).

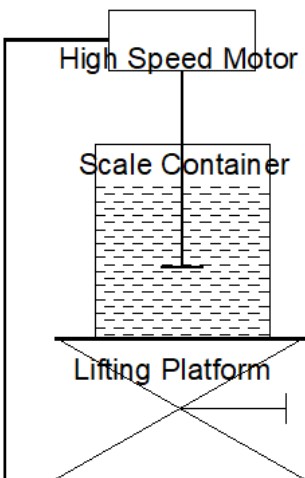

**Figure 1.** Schematic diagram of the preparation device.

### 2.2. Oil Displacement Experiments

The oil displacement experiments with the modified CGAs were carried out with a core flow tester. Two displacement modes were designed in the experiment: one displacement mode was the parallel mode with three sandpack cylinders, and the other displacement mode was the gradient mode with the same sandpack cylinder (the gradient-mode experiments consisted of water flooding first, and when no oil was produced within 2 min during water flooding, CGA flooding was then conducted; after no oil was produced within 2 min during CGA flooding, modified CGA flooding was then carried out). The parameters of the sandpack cylinders in the three systems are listed in Table 1. The experimental conditions included a temperature of 25 °C, a stirring time of 4 min, a stirring

speed of 8000 rpm, a foaming agent SDS concentration of 5 g/L, and a modifier GXJ-C concentration of 0.1 g/L. The experimental oil was a mixture of Venezuela Merey16 (50 wt.%) and Xinjiang heavy oil (50 wt.%), with a density (at 20 °C) of 0.9217 g/cm$^3$, a kinematic viscosity (at 50 °C) of 224.6 mm$^2$/s, and an asphaltene content of 4.87%. The oil was a low-quality crude oil with a high asphaltene content.

**Table 1.** Parameters of the sandpack cylinders in the oil displacement experiments. CGA-colloidal gas aphron.

| Item | Oil Displacement System | Void Volume (mL) | Water Phase Permeability (μm$^2$) | Porosity (%) | Oil Saturation (%) |
|---|---|---|---|---|---|
| Sandpack cylinder 1 | Water flooding | 64.7 | 2.15 | 29.6 | 88.3 |
| Sandpack cylinder 2 | CGA flooding | 64.5 | 2.14 | 29.5 | 88.2 |
| Sandpack cylinder 3 | Modified CGA flooding | 64.6 | 2.13 | 29.6 | 88.3 |

For Table 1, the preparation procedure of the saturated sandpack and the measurement procedures for the sandpack properties were as follows: 100–300 mesh quartz sand was accurately filled to a height of 30.11 cm in a cylinder with a cross-sectional area of 7.25 cm$^2$, and, during filling, the sand column was regularly compacted with a hammer. The cylinder was weighed after completion of quartz sand filling, and then the sandpack was saturated with water in a vacuum until completely saturated. Thereafter, the cylinder was weighed again, and the pore volume and porosity were calculated, while the water injection pressure difference was measured at a speed of 9.8 mL/min. According to the equation, $K_W = Q_W \cdot \mu \cdot L/(A \cdot \Delta P) \times 10^{-9}$, the permeability of the water phase was calculated. Oil was injected into the sandpack cylinder saturated with water, where the volume of displaced water represents the void volume filled with oil, which is the oil saturation.

## 3. Results and Discussion

### 3.1. Selection of the Foaming Agents

To optimize the foaming agent, the selected surfactants included anionic, cationic, and nonionic surfactants. The foaming properties of different CGA systems prepared by adding various modifiers were compared. SDBS, SDS, OP-10, Tween-20, CTAB, and SYHSY were all obtained from Sinopharm Chemical Reagent Co., Ltd. The experiments were conducted at 15 °C, with a stirring time of 4 min and speed of 8000 rpm, to study the foaming volume and half-life period of SDBS, SDS, OP-10, Tween-20, Tween-80, and SYHSY at a concentration of 5 g/L, and the results are shown in Figure 2.

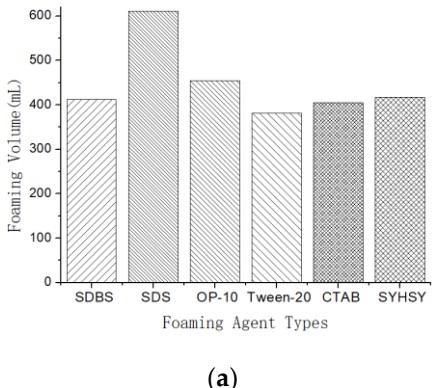
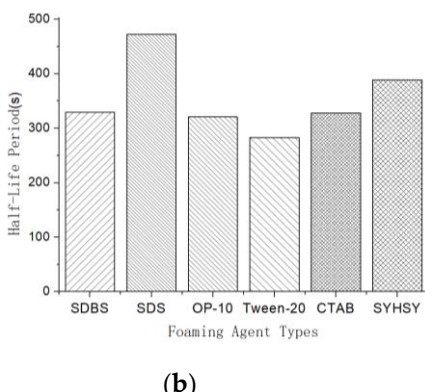

(**a**)                    (**b**)

**Figure 2.** Effects of the different foaming agents on the foam volume and half-life period of the modified CGAs: (**a**) foam volume; (**b**) half-life period.

The results indicated that the addition of SDS solution attained the largest foam volume, almost 580 mL, followed by OP-10 at 455 mL, and the foam volume when using Tween-80 was the smallest at 290 mL. SDS also had the maximum half-life period, almost 9 min. The half-life period of the similar surfactant SYHSY was relatively shorter, close to 7 min, and that of OP-10 and SDBS was approximately 5 min. In this study, validation experiments were also conducted. CGA (610 mL) was

obtained by adding 5 g of SDS foaming agent to 100 mL of water at 15 °C, stirring for 3 min at a speed of 8000 rpm. The bubble diameter range of the system was 0.5–20 μm, and the average diameter was 5.7 μm. The results were basically consistent with the literature [9].

### 3.2. Selection of the Modifier

The modifier selected in the experiment was a low-molecular-weight liquid hydrocarbon, which came from a light fraction of petroleum. The difference between GXJ-A, GXJ-B, and GXJ-C was in their distillation range and hydrocarbon composition. The effects of these three modifiers on the CGA properties were studied in experiments. The experimental conditions were as follows: 15 °C, stirring time of 4 min, speed of 8000 rpm, foaming agent SDS concentration of 5 g/L, and modifier concentration of 0.1 g/L. The results are shown in Figure 3.

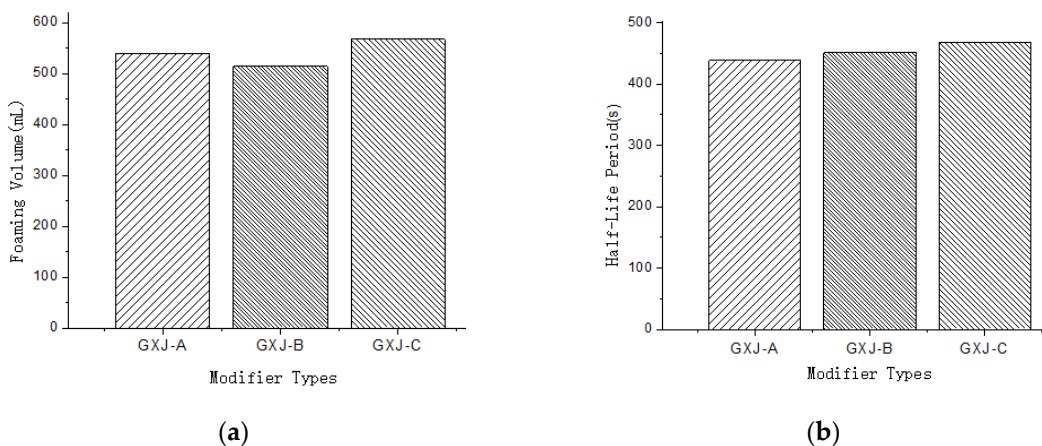

**Figure 3.** Effect of the different modifiers on the foam volume and half-life period of the modified CGAs: (**a**) foam volume; (**b**) half-life period.

The results showed little difference between the effects of the three modifiers on the foam volume and half-life period of the CGAs. Based on the comparison, the modified CGAs prepared with GXJ-C were superior.

### 3.3. Selection of the Modifier Amount

To determine the proper modifier amount, experiments were conducted at 15 °C, with a stirring time of 4 min, a speed of 8000 rpm, and a foaming agent SDS concentration of 5 g/L; the results are shown in Figure 4.

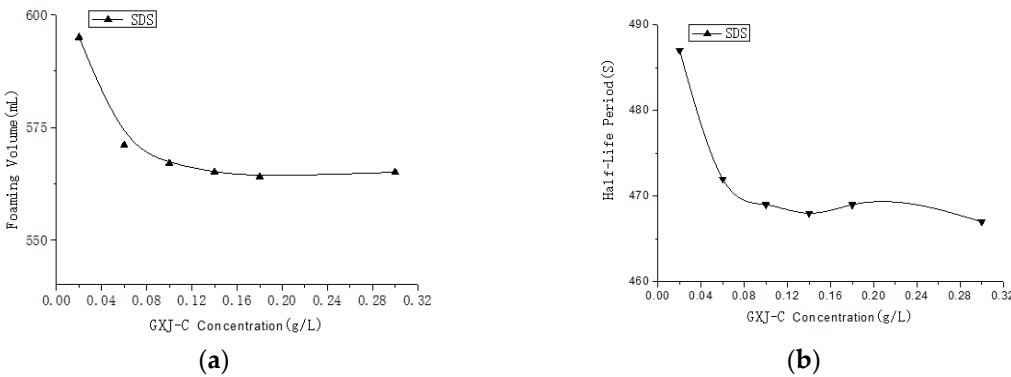

**Figure 4.** Effect of the different concentrations on the foam volume and half-life period of the modified CGAs: (**a**) foam volume and (**b**) half-life period.

The results showed that the foam volume and half-life period of the SDS system firstly decreased and then tended to stabilize with increasing modifier concentration. When the concentration of GXJ-C reached 0.1 g/L, the foam volume and half-life period basically remained stable. The results showed that, when modifier GXJ-C was added to the SDS system, the surface energy of the microbubbles formed in the system slightly increased and the system stability decreased, resulting in decreases in the foam volume and half-life period of the foaming system; however, the decrease range was not large, and the overall effect on the foaming performance of SDS was not notable.

### 3.4. The Effects of Stirring Time, Speed, and Temperature on the Modified CGA Foam

At a modifier GXJ-C concentration of 0.1 g/L and a foaming agent SDS concentration of 5 g/L, the effects of the stirring time, speed, and temperature on the properties of the modified CGA foam were studied, and the results are shown in Figure 5. It should be pointed out that, since the system generated very little foam at stirring speeds lower than 5000 rpm, the conditions at stirring speeds ranging from 5000–10,000 rpm were mainly studied.

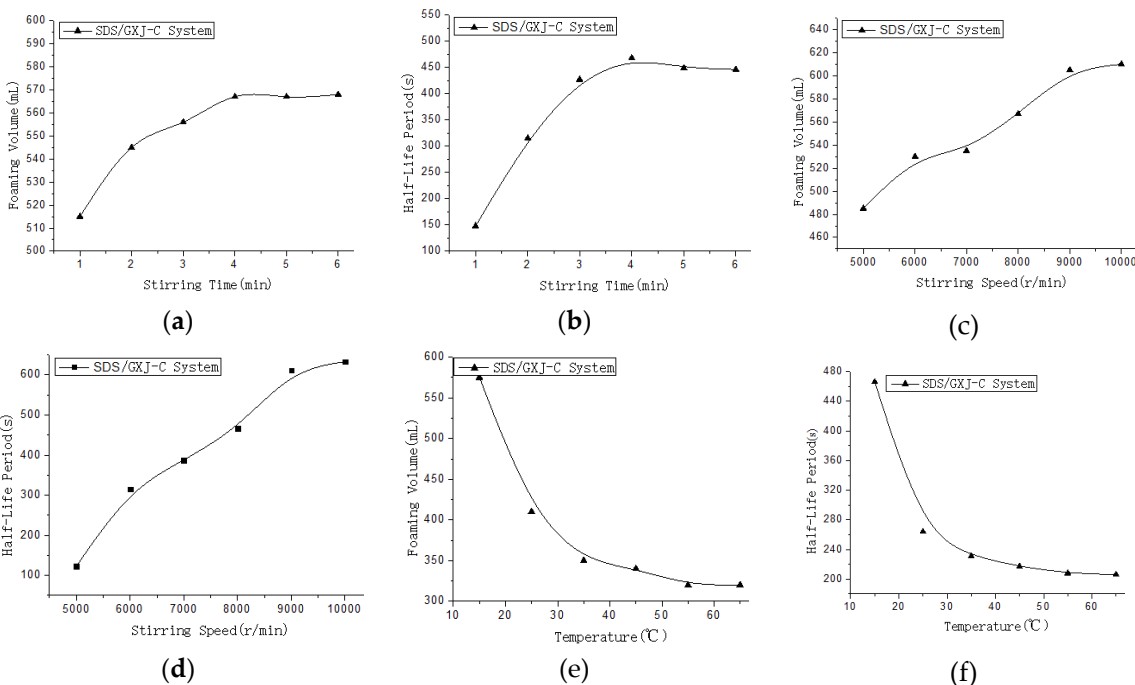

**Figure 5.** Effect of the different preparation conditions on the foaming volume and half-life period of the modified CGAs: (**a**) effect of the stirring time on the foam volume; (**b**) effect of the stirring time on the half-life period; (**c**) effect of the stirring speed on the foam volume; (**d**) effect of the stirring speed on the half-life period; (**e**) effect of the preparation temperature on the foam volume; (**f**) effect of the preparation temperature on the half-life period.

The results showed that the stirring time, speed, and temperature had a large influence on the foaming properties of the modified CGAs, but the influence rules were different. When the stirring time was 3 min, the modified CGA foaming volume and half-life period basically reached the equilibrium state. Within the experimental investigation range, a higher stirring speed led to a larger foam volume and a longer half-life period. When the stirring speed was 9000 rpm, the modified CGA foaming volume basically no longer increased, and the half-life period was no longer extended. When the preparation temperature increased from 15 °C to 35 °C, the modified CGA foaming volume decreased by 40%. For the same half-life period, the modified CGA foaming volume was only 50% of that at a preparation temperature of 15 °C. When the temperature increased, the modified CGA foam volume tended to remain stable with the half-life period.

### 3.5. The Performance of the Modified CGA Foam Flooding

Water flooding, CGA foam flooding, and modified CGA foam flooding were carried out on three sandpack cylinders with similar permeability and oil saturation values (V = 9.5 mL/min). The injection pressure and recovery results are shown in Figure 6.

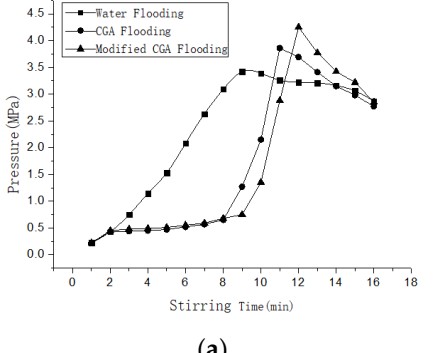
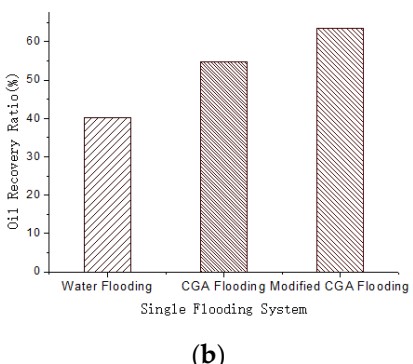

(**a**)    (**b**)

**Figure 6.** The injection pressure and recovery results: (**a**) injection pressure; (**b**) recovery ratio.

The results showed that the injection pressure during water flooding increased over time. After 8 min, the pressure stabilized at approximately 3.1 MPa. Upon increasing oil phase displacement, the pressure gradually decreased; the pressure during CGA flooding and modified CGA flooding reached a peak at 11 and 12 min, respectively, and the oil phase started to be displaced. This phenomenon was related to the pressure accumulation process of foam in the oil zone. The recovery ratio results revealed that the water flooding oil yield was 40.3%, the CGA flooding oil yield was 54.7%, and the modified CGA flooding oil yield reached 63.6%. It is clear that the modified CGA system had a superior recovery effect.

In addition, a series of gradient displacement experiments with the same sandpack cylinder were carried out. That is, water flooding was firstly performed, then CGA flooding, and finally modified CGA flooding. The experimental results are summarized in Table 2.

**Table 2.** Oil recovery ratio in the gradient displacement mode in the three systems with the same sandpack cylinder.

| Parameter | Water Flooding | CGA Flooding | Modified CGA Flooding |
|---|---|---|---|
| Oil recovery ratio (%) | 42.1 | 10.7 | 16.4 |

The data in Table 2 indicate that, for the gradient displacement experiment with the same sandpack cylinder, the recovery ratio was 42.1% by water flooding, 10.7% by CGA flooding, and 16.4% by modified CGA flooding. It can be observed that the CGA flooding recovery was 10.7% on top of that of water flooding, and that the modified CGA flooding recovery was 16.4% relative to that of CGA flooding, indicating that the recovery efficiency of modified CGA flooding was notable.

In summary, the displacement experiments with different sandpack cylinders and the gradient displacement experiment with the same sandpack cylinder all demonstrated that the displacement effect of modified CGA flooding was better than that of water and CGA flooding.

To further clarify the mechanism, experiments were carried out from two aspects: the interfacial tension between the foam system and mixed crude oil, and the content of *n*-heptane asphaltene in gradient flooding with the same sandpack cylinder. The measurement process of the interfacial tension was as follows: in a constant-temperature vessel, crude oil slowly formed droplets through the tip of the tube that fell into the CGA system or modified CGA system. The interfacial tension was calculated according to the equation, $\gamma = f \cdot V \cdot \varrho \cdot g / R$, where V is the measured crude oil volume, $\varrho$ is the crude oil

density, g is the acceleration due to gravity, R is the emitter radius, and f is a correction factor [15]. The asphaltene content was determined with *n*-heptane as the solvent, and, according to the industry standard method, the relative error was within 3%.

The results (Table 3) showed that the interfacial tension of the mixed crude oil and foam system was reduced from 3.29 to 2.68 mN·m$^{-1}$ by adding 0.1 g/L modifier GXJ-C to the CGA system.

**Table 3.** Measurement of the interfacial tension in the CGA-mixed crude oil and modified CGA-mixed crude oil systems.

| Parameter | CGA-Mixed Crude Oil System | Modified CGA-Mixed Crude Oil System |
|---|---|---|
| Interfacial tension (mN·m$^{-1}$) | 3.29 | 2.68 |

Figure 7 shows the asphaltene content data of the crude oil with different system gradient displacements. The figure reveals that the asphaltene content in the oil recovered by water flooding was 4.66%, and that the asphaltene content in the oil recovered by CGA flooding was slightly lower, at 4.44%. However, the asphaltene content in the oil recovered by modified CGA flooding was 5.48%. Compared with the asphaltene content in the mixed crude oil of 4.87%, the changes in asphaltene content in the oil recovered by water flooding, CGA flooding, and modified CGA flooding were −4.31%, −8.83%, and +12.53%, respectively. As the modifier is a crude oil fraction, which is miscible with crude oil, the modified CGA system formed by adding GXJ to CGAs as a modifier clearly reduced the interfacial tension and played a role in dissolving the heavy components in crude oil.

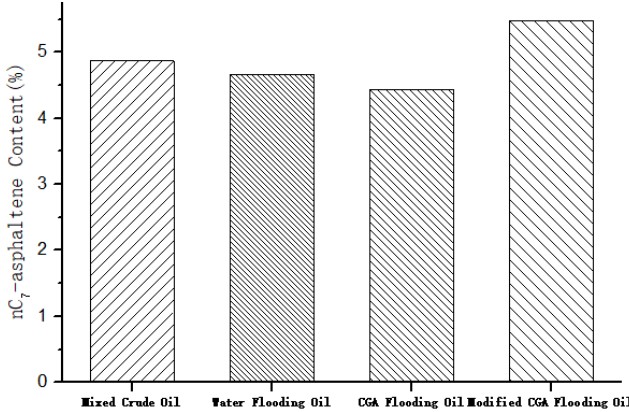

**Figure 7.** *n*C$_7$-Asphaltene content in the oil recovered by the different systems in the gradient displacement mode.

The experimental results showed that the modified CGA system had a good oil displacement effect. On the one hand, it reduced the oil-water interfacial tension, while, on the other hand, it had an elution effect on the asphaltenes deposited in the pores. That is, in the process of CGA flooding, small foam particles firstly entered the large pore channels, and the subsequent displacement fluid was forced to enter the small pore channels due to the foam resistance to displace the oil that could not be reached in water flooding, which enlarged the influenced volume. Compared with conventional CGAs, the modified CGAs reduced the interfacial tension between oil and foam and exerted a certain elution effect on the asphaltenes in crude oil. An emulsified oil zone was more readily formed with a notable oil displacement effect.

## 4. Conclusions

(1) The experimental results showed that the foaming properties of the anionic surfactant SDS were the best, while those of the nonionic surfactant Tween-80 were the worst. Under the experimental conditions, the modified CGA system prepared with foaming agent SDS (concentration: 5 g/L) and

GXJ-C (concentration: 0.1 g/L) had the best properties. A 567-mL CGA system could be prepared from 100 mL of modified solution with a half-life period of 468 s.

(2) The oil displacement experiments showed that the recovery ratio by modified CGA flooding was increased from 40.3% to 63.6%, and it increased by approximately 16.4% in gradient-mode flooding, which indicated that the displacement effect of the modified CGA system was notably better than that of water and CGA flooding.

(3) There were two important mechanisms via which the modified CGAs enhance oil recovery, including decreasing the interfacial tension and enhancing asphaltene elution in crude oil. After adding modifier GXJ-C, the interfacial tension in the CGA-oil system decreased from 3.29 to 2.68 mN·m$^{-1}$, and the asphaltene content in the recovered oil increased by 12.53% compared with that in the mixed crude oil.

**Author Contributions:** S.Y. performed the experiments and initiated the manuscript, X.S. participated in the revision of the manuscript, and X.L. adjusted the format of the manuscript. All authors have read and agreed to the published version of the manuscript.

**Funding:** This research was funded by the Fundamental Research Funds for the Central Universities, grant number 17CX02009A.

**Conflicts of Interest:** The authors declare no conflicts of interest.

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
