# Peer review of "Preparation of Modified Colloidal Gas Aphrons and Analysis of the Oil Displacement Effect"

_applsci, doi:10.3390/app10030927_

Round 1

Reviewer 1 Report

The authors have considered my suggestions and the revised manuscript appears significantly improved. However, the text is still not fluent, as it contains a lot of mistakes and strange phrases; extensive editing is required. Just to list the first examples I found:

Abstract, line 1: "are some advantages" should be "offer some advantages"

Abstract, line 4: "indicate: The" should be "indicate that the"

Abstract, line 4: "by foaming agent" should be "by adding the foaming agent"

Abstract, line 5: delete "system"

This point has to be fixed before publication 

Author Response

Thank you for your comments on our manuscript entitled “Preparation of modified colloidal gas aphrons and experiment of its oil displacement effect” (Manuscript ID: applsci-683792). The comments were valuable and very helpful for improving our paper. We have carefully reviewed the comments and made the recommended revisions, and the revised portions of the paper are indicated in red font. The corrections and the responses to the reviewers’ comments are as follows:

Reviewer 1

Issue 1: Abstract, line 1: "are some advantages" should be "offer some advantages"

Response: Thank you for this suggestion. The "are some advantages" was changed to "offer some advantages" in the revised manuscript (Line 9).

Issue 2: Abstract, line 4: "indicate: The" should be "indicate that the"

Response: Thank you for this suggestion. The "indicate: The" was changed to "indicate that the" in the revised manuscript (Line 12).

Issue 3: Abstract, line 4: "by foaming agent" should be "by adding the foaming agent"

Response: Thank you for this suggestion. The "by foaming agent" was changed to "by adding the foaming agent" in the revised manuscript (Line 12).

Issue 4: Abstract, line 5: delete "system"

Response: Thank you for this suggestion. The "system" was deleted in the revised manuscript (Line 13).

Academic Editor

Issue 1: The article describes the use of colloidal gas aphrons modified by the addition of an unknown agent (GXJ-A, GXJB and GXJ-C) to extract oil from sand packs.

Response: Thank you for this suggestion. The more detailed information about the agents (GXJ-A, GXJB and GXJ-C) are provided in the revised manuscript (lines 111-113 and lines 201-203).

Issue 2: Although most of the comments and suggestions raised by the previous referees have been addressed in the last version of the MS. I will recommend a thorough revision of the English language before continuing to peer-review.

Response: The English level was re-edited by a native-English-speaking editor to ensure that all the grammar/typo errors were corrected. The following editorial certificate was obtained:

Reviewer 2 Report

The manuscript entitled « Preparation of Modified Colloidal Gas Aphrons and Experiment of Its Oil Displacement Effect ” is devoted to the study of colloidal foam system improved by modifiers.

  The article is clearly written, the illustrations are understandable. I believe that this manuscript will be interesting for all scientists involved in such studies and I suggest publication after minor revision of manuscript.

What do you mean by the individual difference of foam in the introduction part? Please rephrase this part. What does “the body water” mean?

 However, at the interface between the gas and water phase, the hydrophobic groups are toward the soap film, and the hydrophilic groups are oriented toward the body water, which have double electric layers. So there is a big difference between the nature and ordinary bubbles[9]

Please improve and rephrase the abstract, as the part of it was just taken from manuscript body.    What do you mean by “The experimental results show that the surface energy of the micro bubbles formed by GXJ-C is increased, which has a certain effect on the foaming performance of SDS, but the overall effect is not significant”. What kind of certain effect? How did you calculate %? “Compared with mixed crude oil of 4.87%, the asphaltene content in oil displaced is increased by -4.31%, -8.83% and +12.53% respectively” How the asphaltene content can have negative value? There are lot of technical mistakes, for example: The asphaltene content of crude oil displaced by gradient in different systems is also measured. as shown in Figure 7. Please check the manuscript again to improve English grammar, I suggest to check it by native speaker, as there a lot of grammar mistakes, for instance, “Adding a certain concentration of foaming agent into” should be “Adding of a certain concentration of foaming agent into” “ The mechanism of foam flooding can be divided into microcosmic displacement of oil and macroscopic displacement of oil” should be changed to “The mechanism of foam flooding can be divided into microcosmic and macroscopic oil displacement”  This study also carried out validation experiments.? and other mistakes

Author Response

Thank you for your comments on our manuscript entitled “Preparation of modified colloidal gas aphrons and experiment of its oil displacement effect” (Manuscript ID: applsci-683792). The comments were valuable and very helpful for improving our paper. We have carefully reviewed the comments and made the recommended revisions, and the revised portions of the paper are indicated in red font. The corrections and the responses to the reviewers’ comments are as follows:

Reviewer 2

Issue 1: Page 1, 1. Introduction, line9: What do you mean by the individual difference of foam in the introduction part?

Response: Thank you for this suggestion. The term "the individual difference" was changed to "the size difference" in the revised manuscript as follows (line 31):

Regarding the size difference of foam and its pseudoplastic effects, when the resistance in the macropores has increased resulting in a higher driving gradient pressure than that in the micropores, the water phase in the foam system may permeate the micropores, and the foam will disintegrate.

Issue 2: Page 2, 1. Introduction, line4-7: Please rephrase this part. What does “the body water” mean?

Response: Thank you for this suggestion. The term "the body water" was changed to "the bulk water" in the revised manuscript as follows (line 52):

However, at the interface between the gas and water phases, the hydrophobic groups are oriented toward the soap film, and the hydrophilic groups are oriented toward the bulk water phase, which results in double electric layers. Therefore, there is a large difference between natural and synthetic bubbles[9].

Issue 3: Page 1, Abstract, lines 1-10: Please improve and rephrase the abstract, as the part of it was just taken from manuscript body.

Response: Thank you for this comment. improved the abstract as follows (Lines 9 to 19):

Colloidal gas aphrons (CGAs) offer some advantages in improving oil recovery, but resin and asphaltene deposition problems still occur in CGA flooding. Based on this phenomenon, a new modified colloidal foam system is developed by incorporating a modifier in CGA preparation. The results indicate that the modified CGAs prepared by adding foaming agent sodium dodecyl sulfate (SDS) (concentration: 5 g/L) and GXJ-C (the modifier; concentration: 0.1 g/ L) attain the best performance. Oil displacement experiments show that modified CGA flooding has a better effect than water or CGA flooding. There are two important mechanisms by which modified CGAs enhance oil recovery, including decreasing the interfacial tension and enhancing the heavy components in the recovered oil. The developed modified CGA system attains a good oil displacement effect, which is of guiding significance to further improve the oil displacement efficiency and application of foam flooding.

Issue 4: Page 4, 3.2. Selection of Modifier Types, lines 13-17: What do you mean by “The experimental results show that the surface energy of the micro bubbles formed by GXJ-C is increased, which has a certain effect on the foaming performance of SDS, but the overall effect is not significant”. What kind of  certain effect?

Response: Thank you for this comment. The term“The experimental results show that the surface energy of the micro bubbles formed by GXJ-C is increased, which has a certain effect on the foaming performance of SDS, but the overall effect is not significant” was changed to " The results show that when modifier GXJ-C is added to the SDS system, the surface energy of the microbubbles formed in the system slightly increases and the system stability decreases, resulting in decreases in the foam volume and half-life period of the foaming system, but the decrease range is not large, and the overall effect on the foaming performance of SDS is not notable." in the revised manuscript as follows (lines 130-134):

The results show that when modifier GXJ-C is added to the SDS system, the surface energy of the microbubbles formed in the system slightly increases and the system stability decreases, resulting in decreases in the foam volume and half-life period of the foaming system, but the decrease range is not large, and the overall effect on the foaming performance of SDS is not notable.

Issue 5: Page 7, 3.5. The Effect of Modified CGA on Oil Displacement, lines 4-6: How did you calculate %? “Compared with mixed crude oil of 4.87%, the asphaltene content in oil displaced is increased by -4.31%, -8.83% and +12.53% respectively”

Response: Thank you for this comment. The term“Compared with mixed crude oil of 4.87%, the asphaltene content in oil displaced is increased by -4.31%, -8.83% and +12.53% respectively” was changed to " Compared with the asphaltene content in the mixed crude oil of 4.87%, the changes in asphaltene content in the oil recovered by water flooding, CGA flooding and modified CGA flooding are -4.31%, -8.83%, +12.53%, respectively." in the revised manuscript as follows (lines 199-201):

Compared with the asphaltene content in the mixed crude oil of 4.87%, the changes in asphaltene content in the oil recovered by water flooding, CGA flooding and modified CGA flooding are -4.31%, -8.83%, +12.53%, respectively. As the modifier is a crude oil fraction, which is miscible with crude oil, the modified CGA system formed by adding GXJ to CGAs as a modifier clearly reduces the interfacial tension and plays a role in dissolving the heavy components in crude oil.

Issue 6: Page 7, The Effect of Modified CGA on Oil Displacement, lines 5-7: How the asphaltene content can have negative value?

Response: Thank you for this comment. The explanation is as follows: The asphaltene content in mixed cude oil is 4.87%. The asphaltene content in oil recovered by water flooding is 4.66%. The asphaltene content in oil recovered by CGA flooding is 4.44%. The asphaltene content in oil recovered by the modified CGA is 5.48%. Compared with mixed crude oil of 4.87%, The change of asphaltene content in oil recovered by water flooding is (4.66%-4.87%)/4.87%*100% = -4.31%,Other analogy. (lines 188-190)

Issue 7: Page 7, 3.5. The Effect of Modified CGA on Oil Displacement, line 1: There are lot of technical mistakes, for example: The asphaltene content of crude oil displaced by gradient in different systems is also measured. as shown in Figure 7.

Response: Thank you for this comment. The term“The asphaltene content of crude oil displaced by gradient in different systems is also measured. as shown in Figure 7” was changed to " Figure 7 shows the asphaltene content data of the crude oil with different system gradient displacements." in the revised manuscript as follows (lines 195-196):

Figure 7 shows the asphaltene content data of the crude oil with different system gradient displacements. The figure reveals that the asphaltene content in the oil recovered by water flooding is 4.66% and that the asphaltene content in the oil recovered by CGA flooding is slightly lower, at 4.44%. However, the asphaltene content in the oil recovered by modified CGA flooding is 5.48%.

Issue 8: Please check the manuscript again to improve English grammar, I suggest to check it by native speaker, as there a lot of grammar mistakes, for instance

Response: Thank you for this proposal. We have asked a native speaker to correct the language and grammar mistakes.

Issue 8: Page 1, 1. Introduction, line 22: - “Adding a certain concentration of foaming agent into” should be “Adding of a certain concentration of foaming agent into”

Response: Thank you for this comment. The term“Adding a certain concentration of foaming agent into” was changed to "Adding of a certain concentration of foaming agent into" in the revised manuscript as follows (line 25):

Adding of a certain concentration of foaming agent into the water is supplemented by gas (air, natural gas or flue gas) to form foam, and its high viscosity helps reduce the mobility ratio and increase the sweep efficiency of the injectant[1,2].

Issue 8: Page 1, 1. Introduction, lines 24-25: -“The mechanism of foam flooding can be divided into microcosmic displacement of oil and macroscopic displacement of oil” should be changed to “The mechanism of foam flooding can be divided into microcosmic and macroscopic oil displacement” and other mistakes

Response: Thank you for this comment. The term“The mechanism of foam flooding can be divided into microcosmic displacement of oil and macroscopic displacement of oil” was changed to "The mechanism of foam flooding can be divided into microcosmic and macroscopic oil displacement" in the revised manuscript as follows (lines 27-28):

The mechanism of foam flooding can be divided into microcosmic and macroscopic oil displacement [3,4].

Issue 8: - This study also carried out validation experiments.?

Response: This study does have validation experiments.

Academic Editor

Issue 1: The article describes the use of colloidal gas aphrons modified by the addition of an unknown agent (GXJ-A, GXJB and GXJ-C) to extract oil from sand packs.

Response: Thank you for this suggestion. The more detailed information about the agents (GXJ-A, GXJB and GXJ-C) are provided in the revised manuscript (lines 111-113 and lines 201-203).

Issue 2: Although most of the comments and suggestions raised by the previous referees have been addressed in the last version of the MS. I will recommend a thorough revision of the English language before continuing to peer-review.

Response: The English level was re-edited by a native-English-speaking editor to ensure that all the grammar/typo errors were corrected. The following editorial certificate was obtained:

This manuscript is a resubmission of an earlier submission. The following is a list of the peer review reports and author responses from that submission.

Round 1

Reviewer 1 Report

Preparation of Modified Colloidal Gas Aphrons and Experiment of Its Oil Displacement Effect

The article describes the use of colloïdal gas aphrons modified by the addition of an unknown agent (GXJ-A, GXJB and GXJ-C) to extract oil from sandpacks.

Despite the work that has been done, the article has some major flaws

Please find a list of some of the problems that have to be addressed before resubmission of this paper. I hope it will help the authors to improve their document.

Materials and methods:

What are GXJ-A, GXJ-B and GXJ-C? Where did you get those products?

Chemicals provenance is not listed.

The preparation procedure of the saturated sand pack is not explained.

The procedure for the measure of the sand pack properties is not given.

Foaming volume and half-life (Figure 2): very basic characterisation of the foam, no error bars, no characterization of the foam bubbles size, no reproducibility?

How is the interfacial tension measure between the mixed crude oil and the foam system?

Figure 7 How is the asphaltene content evaluated?

There are some typographical, conjugation and grammatical errors, including for example:

P1 Gernerally

P1 For the differences among foams and its pseudoplastic

P2 The preparation steps of modified CGA are as following

P6 and the content of n-heptane asphaltene content

P6 the asphaltene content in displacement oil is increased

Some scientific errors

Colloïdal gas aphrons are made of microbubbles encapsulated by surfactant multilayers and not particles as in the first line of the abstract and at the end of first page

Some descriptions are unnecessary:

p2 The preparation of the modified CGA : “The preparation steps of modified CGA are as following: (1) Weighing 4g foaming agent, add to the 1000mL volumetric flask; (2) Weighing 0.1g modifier, add to the same volumetric flask; (3) Add distilled water to the volumetric flask to the scale line and shake it until foaming agent is completely dissolved

Some expressions are not scientific or should be avoided,

p3 SDS has the largest foam

p6 The experimental results show that the modified CGA flooding effect is good

p6 Modified CGA 567mL can be prepared by 100mL liquid, in 468s at half-life.

P7 modified CGA flooding is obviously better

Figures:

P2 what is the purpose of including figure 2?

P There is an error on Figure 5b: Stirring Time (r/min) it is either stirring rate or stirring time (min)

P4 Figure 3. Effects of different foaming agents on the foaming volume and half-life period of modified this is the description of Figure 2 and not figure 3

P4 Figure 4. Effects of different foaming agents on the foaming volume and half-life period of modified this is the description of Figure 2 and not figure 4

Figure 3 effect of the modifier, no error bar end the difference between the modifier is not really significant.

Figure 4 no error bars, the figure 4b need more comment in the text

Figure 5 I don’t understand what is the point of these figures.

Figure 6 there is no error bars for my point of view the difference between CGA and modified CGA is due to the sand bag

P2 Table1: how are those parameters measured?

What is the procedure used to saturate the sandpacks (aging?)

The authors ignore much of the literature on the subject, for exemple F. Sebba

(J. Colloid Interface Sci., 35 (1971), pp. 643-646) first description of the aphrons gas. Some of the first descriptions of aphrons for oil recovery (Bjorndalen N, Kuru E. Physico-chemical characterization of aphron based drilling fluids. J Petro Sci Eng. 2008;47:15–21, Bjorndalen H, Jossy W, Alvarez J, Kuru E. A laboratory investigation of the factors controlling the filtration loss when drilling with colloidal gas aphron (CGA) fluids. J Petro Sci Eng. 2014;117:1–7. doi: 10.1016/j.petrol.2014.03.003. Brookey T (1998) Micro-bubbles: New aphron drill-in fluid technique reduces formation damage in horizontal wells. SPE 39589 presented at International Symposium on Formation Damage Control, Lafayette, Louisiana, 18–19)

The references given by the authors are not accessible (ref4) or I can’t find them (ref2 and 3) maybe are they uniquely published in Chinese. There are also spelling mistakes like in ref 12.

Reviewer 2 Report

This manuscript reports the preparation procedure of a modified colloidal Gas Aphrons to be used for enhanced oil recovery. The composition and the procedure is optimized and some preliminary tests of functionality are presented. A few physico-chemical data of interfacial tension are also reported. The system is quite interesting, in consideration of the urgent need of effective agents to enhance oil recovery. In principle, the work could be published on applied science. However, the paper seems, in most of the section, a technical report more rather than a scientific contribution. Moreover, very specialized terms are used and not explained, and this reduces its appealing for a large readership.

I would suggest the authors to enlarge and strengthen the work under these aspects. The text is also written in a very bad English and must be checked and emended before it could be published.

Specific points:

- The abstract fails in showing the relevance of the study. It should start with a phrase clarifying the field of interest. Acronyms of chemical substances have to be explained.

- In the introduction the definition of “colloidal gas” should be discussed, since it is not trivial.

- The chemical composition of the standard CGA should be better detailed.

- Details of the used stirrer have to be reported

- The gradient mode experiments should be better explained in the Method section.

- Some interpretations of the effects of the different surfactants shown in Figure 2 should be proposed.

- What are, from a chemical point of view the GXJ modifiers? Which is their mechanism of action?

- The modifiers reduce the foaming volume and the half-life period (Figure 4). This appears in contrast wit their enhanced functionality. This has to be better discussed by the authors.

- The uncertainties on all the measured quantities have to be quoted. In some cases, I wonder if commented differences are really significant (e.g., Figure 7)

- Experimental details of interfacial tension measurements should be reported.